# DREAM:
# Deployment of Recombination and Ensembles in Argument Mining

**Florian Ruosch** and **Cristina Sarasua** and **Abraham Bernstein**
University of Zurich, Switzerland
`lastname@ifi.uzh.ch`

## Abstract

Current approaches to Argument Mining (AM) tend to take a holistic or black-box view of the overall pipeline. This paper, in contrast, aims to provide a solution to achieve increased performance based on current components instead of independent all-new solutions. To that end, it presents the **D**eployment of **R**ecombination and **E**nsemble methods for **A**rgument **M**iners (**DREAM**) framework that allows for the (automated) combination of AM components. Using ensemble methods, DREAM combines sets of AM systems to improve accuracy for the four tasks in the AM pipeline. Furthermore, it leverages recombination by using different argument miners elements throughout the pipeline. Experiments with five systems previously included in a benchmark show that the systems combined with **DREAM** can outperform the previous best single systems in terms of accuracy measured by an AM benchmark.

## 1 Introduction

A well-known and open challenge in Argument Mining (AM) is that approaches do not generalize well across domains (Lippi and Torroni, 2016a). Thus, a single system will not be able to solve the task of extracting arguments from publications across multiple research fields. Therefore, we investigate the use of ensemble methods (Opitz and Maclin, 1999) to find combinations that are expected to help alleviate the issue.

Furthermore, the overview of AM approaches presented by Lawrence and Reed (2019) shows that papers typically introduce novel techniques or use methods for AM that have demonstrated success in other applications. As new systems tend to take the holistic view of an end-to-end pipeline (Lawrence and Reed, 2019), it has become evident that novel approaches rarely investigate improvements of intermediate steps. By introducing a system with ensemble methods and combinations, we aim to improve smaller aspects of the pipeline.

Moreover, Lawrence and Reed (2019) also take the same line by advocating for a unifying framework to enable the harmonization of all AM tasks, including the format of data and results. Such a unification would be necessary to combine many systems and facilitate the integration of additional ones. Likewise, not every task receives the same amount of attention, with approaches for identifying argumentative relations being sparse (Al-Khatib et al., 2021). Thus, they cover a smaller range of domains or do not work well across them. By using recombination, we hypothesize to increase coverage and find yet untapped potential.

To this end, we formulate the following research question:

**RQ.** How can we leverage (re-)combinations of Argument Mining systems to improve accuracy?

Thus, we build **DREAM**, a system that allows for the **D**eployment of **R**ecombination and **E**nsemble methods for **A**rgument **M**iners. For this endeavor, we base our approach and the evaluation on BAM (Ruosch et al., 2022), our benchmark for Argument Mining. We reuse the performance data of five AM systems when evaluated by BAM as well as its implementation for our purposes. Accordingly, we restrict the systems for the initial combinations to these five argument miners and adhere to the definition of the four tasks in the AM pipeline (Lippi and Torroni, 2016a): *sentence classification*, *boundary detection*, *component identification*, and *relation prediction*. Using these tools, we try to outperform the current best accuracy for every task of the AM pipeline by combining systems with the following ensemble methods: *voting*, *stacking*, and *bagging*. Finally, we split the AM systems into "modules" according to the AM pipeline, allowing their recombination to increase accuracy.

We present two main contributions in this paper. First and foremost, we build the **DREAM** framework to combine AM systems using ensemble

methods and recombinations. Second, we show the value of such combinations, as they outperform some of the state-of-the-art systems used in the AM benchmark.

The remainder of this paper is structured as follows: Section 2 presents background and the related work, and Section 3 introduces our methodology. In the ensuing Section 4, we describe our experiments and their results before we evaluate them in Section 5. Then, Section 6 discusses limitations and future work. Finally, we draw conclusions in Section 7.

## 2 Background

In this section, we lay the foundations by describing Argument Mining and presenting specific related work.

### 2.1 Argument Mining

The field of AM is wide-ranging and has different interpretations (Wells, 2014) of what its tasks consist of. We focus on the information extraction approach (Budzynska and Villata, 2015; Lippi and Torroni, 2016a): the automated analysis of arguments in natural language text. To this endavor, we consider the AM pipeline as described by Lippi and Torroni (2016a), depicted in Figure 1. The input text is processed in four stages: *argumentative sentence detection*, *argument component boundary detection*, *argument component detection*, and *argument structure prediction*.

In the first step, sentences are classified as argumentative if they contain parts of an argument and as non-argumentative otherwise. Next, the boundaries of the argument components are identified by segmenting the argumentative sentences. Then, these argument components are classified according to the representation of the arguments defined beforehand. Finally, the structure (i.e., relations) of the previously identified components is predicted to form an argument graph. The annotated text (in any format) is the output of the AM pipeline.

### 2.2 Specific Related Work

Combining approaches in AM has barely received any attention in previous literature. The only exception is the work of Lawrence and Reed (2015), where the authors implement and combine three different AM techniques. They are evaluated with respect to identified connections between propositions and use a fixed set of 36 pairs.

First, the presence of *discourse indicators*: words such as "because" and "however", indicating *support*- and *conflict*-relations, respectively, between adjacent statements. These words provide a good signal (precision of $1.00$), but the technique fails to capture most relations (recall of $0.08$) due to their low number of occurrences in texts. Furthermore, they can not be used to find relations for non-adjacent propositions.

The second technique is based on changes in the topic for consecutive propositions, which is assumed to relate to the argumentative structure in the text. The similarity of adjacent propositions is calculated using the synsets of WordNet[1], resulting in a number on a scale from 0 to 1. A preset threshold then determines whether the topic remains the same, and, that being the case, it is deduced that the propositions are connected. This approach achieves a precision of $0.70$ and a recall of $0.54$, respectively.

The third method uses argumentation schemes (Walton et al., 2008): "common patterns of human reasoning." They avoid the need for having the components and the structure of arguments already annotated by instead focusing on features of the parts of the present scheme. With a list of propositions from the text and a Naïve Bayes classifier, they can determine the particular scheme and, therefore, detect information about the structure of the argumentation. This results in a precision of $0.82$ and a recall of $0.69$.

Finally, the techniques are combined to exploit their respective fortes. The presence of discourse indicators is used to infer connections among propositions in the first step. Subsequently, components are related after having determined argumentation scheme instances. Lastly, previously unconnected units are integrated based on topic similarity. Combining the methods results in an improved performance with precision and recall, increasing to $0.91$ and $0.77$, respectively.

In contrast to the approach described above, we aim to provide combinations on a larger scale and a pipeline for a unifying framework that allows for integrating additional components. We aim to investigate if and how combinations (of parts of) different AM system can be used to improve overall performance. Finally, our approach also differs

---

[1] http://wordnet.princeton.edu

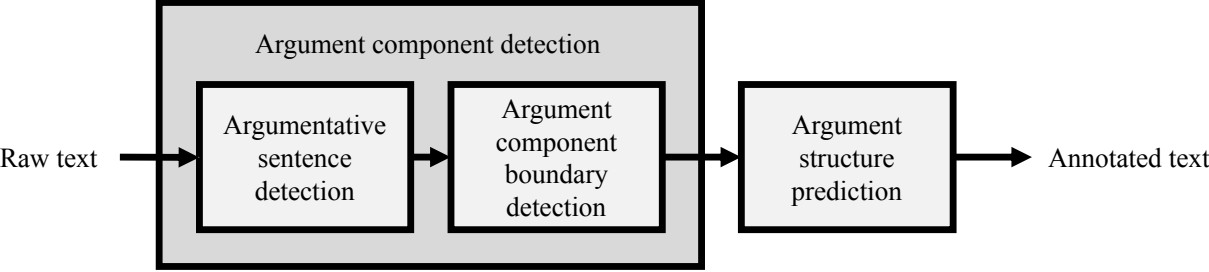

Figure 1: The Argument Mining pipeline adapted from Lippi and Torroni (2016a).

in that we do not look to combine techniques or features but rather out-of-the-box argument miners. This facilitates the integration of additional systems.

## 3 Methodology

In this section, we first lay out our evaluation hypotheses and then describe the concept of the **DREAM** (re-)combination framework.

### 3.1 Hypotheses

We base both the evaluation and the conception on the systems, data, and results of BAM, our benchmark for Argument Mining (Ruosch et al., 2022). We utilize the five systems included in BAM (i.e., AURC, TARGER, TRABAM, ArguminSci, and MARGOT) and evaluate the recombinations using the provided implementation of the benchmark. That means the AM pipeline is split into four tasks (Lippi and Torroni, 2016a): *sentence classification*, *boundary detection*, *component identification*, and *relation prediction*. These tasks are evaluated with their respective metrics from BAM (i.e., micro F1, the boundary similarity measure defined by Fournier (2013), and F1-score).

To evaluate the implemented recombination system, we formulate the following hypotheses derived from the research question and describe our approach to assess their acceptance or rejection.

**H1.** For some tasks in the AM pipeline, ensembles of systems exist for which accuracy will be higher than for the most accurate single system.

This hypothesis encapsulates two different but entangled problems: finding the optimal set of systems to combine and testing whether they are more accurate than the current top system. Thus, we split it into two sub-hypotheses, which are the requisites for accepting Hypothesis **H1**.

**H1.1.** There exists an ensemble of systems for every task that is more accurate than any other ensemble of systems (excluding single systems).

Since we already restrict the space of systems and combinations that we need to explore by limiting ourselves to the systems in BAM, we can test all combinations of size $n$, where $1 < n \leq 5$, because we require combinations of at least two and can combine at most all five systems. Thus, it becomes a matter of running all possible systems and combining them using ensemble methods We accept the hypothesis if we find one or more ensembles of systems that exceed all others in terms of accuracy as measured by BAM for all the tasks. It is important to note that these ensembles might differ for individual tasks.

**H1.2.** For some tasks in the AM pipeline, the most accurate ensemble of systems will be more accurate than the most accurate single system for this task.

For the second sub-hypothesis, we can compare the previously discovered combinations with the most accurate single system and compare their numbers for all the tasks. That means doing a pair-wise comparison four times, namely once for every task, and checking whether the combinations outperform the single systems. Again, we accept the hypothesis if we can confirm this for at least some of the four tasks.

**H2.** For some tasks in the AM pipeline, the accuracy for subsequent tasks will be higher if intermediate data is used that has been produced by the system with the highest accuracy for the preceding task instead of its own intermediate data.

Subsequently, we investigate how to improve in single tasks and how the intermediate results influence the ensuing tasks of the pipeline. Thus, we

hypothesize that data of higher accuracy compared to the ground truth will also result in an increase in a system's accuracy as opposed to its own intermediate results. Again, we try out all possible pairs to answer this hypothesis. Any two systems can be combined by using one's output as the other one's input, provided that the former's accuracy was higher than the latter's at the preceding task. Considering the five systems and four tasks in the benchmark, we have to try out a maximum of 60 pairwise combinations, every system acting as "input provider" and "input taker" but never at the same time. We compare the new highest accuracy for every task and system to the previous results and accept this hypothesis if the new numbers are higher than the old ones.

**H3.** For some tasks in the AM pipeline, the accuracy will be higher if we use an ensemble of systems and intermediate results as input produced by the most accurate system (ensemble) for the preceding task.

The final hypothesis brings all possible combinations together. We not only allow combining systems for tasks but also to "mix-and-match" for the intermediate results in the hope of improving the accuracy of the whole pipeline. We employ the best combinations from Hypothesis **H1** and combine them with the insights from Hypothesis **H2**. We compare the newly obtained accuracies to the previous best per the benchmark and accept the hypothesis if we outperform the top single system for every task.

## 3.2 The DREAM Framework

The basic idea behind the approach to combining multiple AM systems is simple: Employ a multitude of systems such that they can combine their strengths and, at the same time, balance out their weaknesses. Our framework, **DREAM**, is intended for the (automated) recombination of multiple AM systems according to predefined parameters. Following the aforementioned AM pipeline by Lippi and Torroni (2016a), we first identify argumentative sentences, then we identify the boundaries of the components and classify them (usually as either *claim* or *premise*). Finally, we predict the relations between the argumentative components (such as *supports* or *attacks*).

Not every argument miner adheres to this pipeline, which results in some of the argument miners lacking the capabilities to solve one or more of these tasks. Furthermore, Lawrence and Reed (2019) point out that current systems tend to take a holistic view of the end-to-end pipeline. This is further emphasized by the fact that black-box models, such as neural networks and, more specifically, transformers, become increasingly prevalent. While they carry the advantage of improved performance, they prevent a look into their inner workings and modularization of their features.

Thus, we have access solely to the final outputs of argument miners for our framework. However, as we showed in BAM (Ruosch et al., 2022), we can reconstruct the intermediate results necessary for evaluating the tasks mentioned above of the AM pipeline. We can use these reconstructed intermediate results for the recombination effort, with the added benefit of not needing to re-train or re-run any of the systems (i.e., we only perform post-hoc combinations).

**DREAM** reads the output files from the argument miners and calculates the combinations according to the specified parameters. There are several different options when combining this data: the list of employed systems, the method to calculate the combination, and the targeted task.

Figure 2 visualizes the ways we combine systems. Figure 2a corresponds to what is described in Hypothesis **H1**: using ensemble methods to combine multiple systems for a single task. This is what we call *Vertical Integration*. Meanwhile, Figure 2b illustrates Hypothesis **H2**: using different systems throughout the AM pipeline (recombination). This is referred to as *Horizontal Integration*. Tying these two together, we get the *Combined Integration*, where we allow sets of systems to be used for the intermediate results fed forward in the pipeline to either other combinations or single systems.

### 3.2.1 Vertical Integration

*Vertical Integration* gets its name because we choose systems from the "column" of options as illustrated in Figure 2a. The number of systems used for the combination can vary from a minimum of two to all available systems. The list of used systems can be either specified or the recombination framework can try (all) possible combinations (including power sets). This is how we approach auto-experimentation for recombination.

As for the method to calculate the combination of results, we follow the well-established ensemble

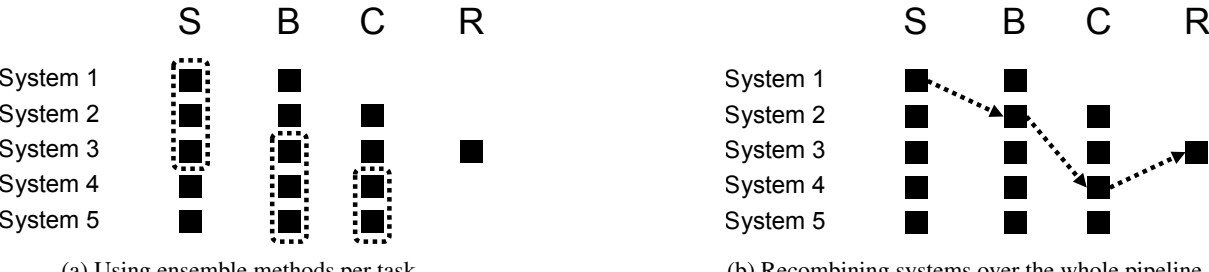

(a) Using ensemble methods per task.    (b) Recombining systems over the whole pipeline.

Figure 2: The two types of combinations employed in the framework.

methods (Opitz and Maclin, 1999). As simplest method, we employ a hard voting scheme (Little-stone and Warmuth, 1994), where systems can op-tionally be assigned a weight. It is important to note that systems may all receive a uniform weight of one or be assigned arbitrary scores (e.g., bench-mark results). Then, we calculate the score of the available answers for a given item based on the systems' output and weights. Next, we use the ensemble stacking method, which trains a meta-classifier on the predictions the trained argument miners produced. We employ multinomial logistic regression (Greene, 2003) as the stacking model. Our third ensemble method is bootstrap aggregat-ing (bagging) (Breiman, 1996). Using bootstrapped sets for training, we expect to strengthen the en-semble of classifiers.

### 3.2.2 Horizontal Integration

Next, we allow combining systems across the bor-ders of individual tasks (i.e., columns) by using intermediate results and feeding them to other sys-tems. This results in what we call *Horizontal Inte-gration*, since we allow the combination of differ-ent "rows", as depicted in Figure 2b. The output for all the tasks in the AM pipeline depends on the input fed into the corresponding module. Although, these modules may not be explicitly constructed as such and may have to be inferred due to the holistic view of AM systems (Lawrence and Reed, 2019). Still, we can generally describe the data processing in the AM pipeline. In the first step, the raw text supplied to the pipeline is split into sentences, which in turn are classified as either ar-gumentative or non-argumentative, depending on the presence of argumentative components in them. Thus, the output of the sentence detection depends on its input because it will process (and output) no more and no less than the text it has been sup-plied with. Subsequently, the boundary detection will find the delineations of components in only the

argumentative sentences since, by definition, only they may contain argumentative components. The same holds for the component identification: it will only identify components whose boundaries have been detected. Lastly, the relation prediction relies on the previously identified components to find the triples (subject and object are from the set of ar-gumentative components) that constitute its output. Thus, we can see that every subsequent step in the AM pipeline depends on its predecessor's output.

### 3.2.3 Combined Integration

Finally, we will also allow for the *Vertically Inte-grated* ensemble learners to be used as the inter-mediate result creators and, thus, bring it together with *Horizontal Integration* to the *Combined Inte-gration*. Since we hypothesize that both individual *Integrations* increase accuracy, we hypothesized their combination exhibits an even higher perfor-mance. Thus, we make an effort to find sequences of combined AM systems that further improve the accuracy of the tasks in the pipeline.

## 4 Experiments

In this section, we discuss conducted experiments. First, we describe the setup used. Then, we ex-plain the implementation of the experiments and the subsequent evaluation.

### 4.1 Setup

We rely on the systems and data used by BAM (Ru-osch et al., 2022), the results of which are shown in Table 1. Thus, we consider five different AM systems that have been benchmarked using the Sci-Arg data set (Lauscher et al., 2018b). It represents the only available collection of full argument an-notated scientific papers in English and builds on the Dr. Inventor data set (Fisas et al., 2016). The corpus consists of publications from the field of computer graphics and contains a total of 10,780

| System | | S | B | C | R |
|---|---|---|---|---|---|
| AURC | (Trautmann et al., 2020) | *0.792* | 0.470 | - | - |
| TARGER | (Chernodub et al., 2019) | 0.653 | *0.483* | *0.656* | - |
| TRABAM | (Mayer et al., 2020) | **0.832** | **0.506** | **0.662** | **0.021** |
| ArguminSci | (Lauscher et al., 2018a) | 0.600 | 0.115 | 0.091 | - |
| MARGOT | (Lippi and Torroni, 2016b) | 0.454 | 0.097 | 0.133 | - |

Table 1: Results of BAM, our benchmark for Argument Mining (Ruosch et al., 2022).

sentences that have been annotated with argumentative components (*background* and *own claims* as well as *data*) and relations (*contradicts*, *supports*, *semantically same*, and *part of*).

BAM uses individual evaluation measures for each of the pipeline tasks. For the argumentative sentence classification, it employs the micro-F1 (van Rijsbergen, 1979) score to avoid the skewing effect of a possible label imbalance. For the boundary detection task, BAM uses the boundary similarity measure as proposed by Fournier (2013), which compares the identified boundaries for two segmentations for the same text. The argumentative component identification is evaluated by using the F1 as implemented for the task of Named Entity Recognition (Segura-Bedmar et al., 2013). Finally, relation prediction is treated as the classification of triples (subject, predicate, object) into retrieved or missed and thus, BAM employs the F1-score. Therefore, we obtain four individual scores between 0 and 1, one per task in the AM pipeline, where bigger signifies better.

We use the same five systems that have already been evaluated in the initial showcase of BAM. The first three were trained on the benchmark data set, while the last two were already pre-trained by the authors of the systems. AURC (Trautmann et al., 2020) treats AM as a sequence tagging problem and employs the BiLSTM model of Reimers et al. (2019) to identify argumentative spans in texts. TARGER (Chernodub et al., 2019) also uses a BiLSTM in conjunction with a CNN-CRF and pre-computed word embeddings to label tokens from free text input as belonging to either *claims* or *premises*. TRABAM (Mayer et al., 2020) relies on pre-trained transformers such as SciBERT (Beltagy et al., 2019) in combination with neural networks. TRABAM is the sole system in the benchmark that solves all the pipeline tasks, tagging argumentative components and predicting relations between them. ArguminSci (Lauscher et al., 2018a) was trained on the data set that is also incorporated in

the benchmark. It consists of a range of different tools to analyze rhetorical aspects, but we only use the argument component identification functionality. This module uses a BiLSTM to tag tokens as one of three argumentative component types akin to the annotations in the corpus: *background claim*, *own claim*, or *data*. Finally, MARGOT (Lippi and Torroni, 2016b) detects *claims* and *evidences* by analyzing the sentence structures and uses a subset tree kernel (Collins and Duffy, 2002) to compare their constituency parse trees.

Data and code involved in the execution and subsequent evaluation are available in the project's repository.[2]

## 4.2 Vertical Integration

The best results for each task using the *Vertical Integration* are presented in Table 2.[3] For context, we also report the runner-up and the worst result, as indicated in the *Result* column, and provide the mean, median, and variance for each task. We round all results to three decimal places for readability, except where necessary to indicate differences. Each task of the AM pipeline is represented by a row, in which the accuracy (as measured by BAM), the used ensemble method, and the systems involved are indicated (the order matters therein as the first system serves as the primary to which all other annotations are aligned to). Notably, the relation prediction score $R$ is absent since only one system performed it in BAM, and thus, there is no opportunity to apply an ensemble method. Also, because no system explicitly disentangles the AM pipeline into individual tasks, we perform the combination on the final output and not on task-specific annotations, akin to the way it is handled in BAM. We tried every possible combination of all system lists and ensemble methods to obtain the results and list the best, second best, and worst here.

---

[2] https://gitlab.ifi.uzh.ch/DDIS-Public/DREAM
[3] The full results are omitted for brevity and are available in the online repository.

| Task | | Result | Score | Method | Systems |
|---|---|---|---|---|---|
| **S** | mean: 0.739 | **Best** | **0.8419** | Stacking | TARGER, AURC, MARGOT, TRABAM |
| | median: 0.793 | Second | 0.8416 | Stacking | TARGER, AURC, TRABAM |
| | variance: 0.011 | Worst | 0.513 | Voting | MARGOT, TARGER |
| **B** | mean: 0.379 | **Best** | 0.4972 | Bagging | TARGER, ArguminSci, MARGOT, TRABAM |
| | median: 0.457 | Second | 0.4971 | Bagging | ArguminSci, MARGOT, TARGER, TRABAM |
| | variance: 0.020 | Worst | 0.022 | Voting | MARGOT, ArguminSci |
| **C** | mean: 0.498 | **Best** | **0.673** | Voting | TRABAM, TARGER |
| | median: 0.615 | Second | 0.671 | Voting | TRABAM, ArguminSci, TARGER |
| | variance: 0.036 | Worst | 0.052 | Voting | MARGOT, ArguminSci |

Table 2: Best, runner-up, and worst results per task for *Vertical Integration* (mean, median, and variance refer to all results per task).[3]

| From | To | B | C | R |
|---|---|---|---|---|
| TRABAM | AURC | **0.475** | - | - |
| TRABAM | TARGER | **0.494** | 0.630 | - |
| TRABAM | ArguminSci | **0.281** | 0.345 | - |
| TRABAM | MARGOT | **0.171** | 0.162 | - |

Table 3: Results per system for *Horizontal Integration*.

Stacking the systems TARGER, AURC, MAR-GOT, and TRABAM using logistic regression is the most accurate ensemble for sentence classification with $S = 0.8419$. Bagging with TARGER, ArguminSci, MARGOT, and TRABAM achieves a score of $B = 0.4972$ for boundary detection, which is the highest among the ensembles. Combining the two systems TRABAM and TARGER using the hard voting scheme results in $C = 0.673$ as the best score for component identification.

The main insight gained from these results is that no ensemble method outperforms the others. Rather, each of the three techniques achieves the highest score for one task.

### 4.3 Horizontal Integration

Table 3 shows the complete results for the *Horizontal Integration*. We used the most accurate system from BAM, TRABAM, as listed in the "From"-column to indicate where the intermediate results originated from. These were combined with the output of the individual systems (in the "To"-column) in the respective rows by using them as the template for the subsequent annotations.

TARGER combined with TRABAM scores the highest for both the boundary detection $B = 0.494$ and component identification $C = 0.630$. Again, due to the lack of a system to combine TRABAM with, the results for the relation prediction $R$ are

omitted. The sentence classification is not considered for the *Vertical Integration* as its input is the initial text, which is not considered an intermediate result since it is the same for every system.

### 4.4 Combined Integration

In Table 4, we show the results of the *Combined Integration*. We list the results achieved with the previously identified most accurate single system or ensemble (from the *Vertical Integration*) and their score for each AM pipeline task. For each row in the table, the output has been combined with the output of the preceding row, according to the *Horizontal Integration*. This results in the *Combined Integration*.

The ensemble of TARGER, AURC, MARGOT, and TRABAM stacked using logistic regression is the most accurate for sentence classification with $S = 0.842$. The single system TRABAM achieves the highest boundary detection score with $B = 0.483$. Combining TRABAM and TARGER into an ensemble using voting results in $C = 0.673$ as the best score for component identification. Finally, TRABAM scores $R = 0.019$ for the relation prediction. Interestingly, ensembles are only better than single systems in two out of the three AM pipeline tasks (relation prediction does not have an alternative to TRABAM).

## 5 Hypotheses Evaluation

In this section, we evaluate the hypotheses individually. The results from BAM in Table 1 serve as the baseline, more specifically, the best-performing system nicknamed TRABAM in with the bold-faced numbers. It achieved the following scores for the AM tasks, where each of them is on a scale from zero to one, and higher means better: sen-

| Task | Score | Method | System(s) |
|:---:|:---:|:---:|:---|
| **S** | **0.842** | Stacking | TARGER, AURC, MARGOT, TRABAM |
| **B** | 0.483 | Single | TRABAM |
| **C** | **0.673** | Voting | TRABAM, TARGER |
| **R** | 0.019 | Single | TRABAM |

Table 4: Results per task for *Combined Integration*.

tence classification $S = 0.832$, boundary detection $B = 0.506$, component identification $C = 0.662$, and relation prediction $R = 0.021$. Unlike the result reported in BAM, we use TRABAM's intermediate results as input for the last step, decreasing the accuracy (from $R = 0.318$ when using the ground truth components). We compare the newly obtained scores to these numbers to evaluate the hypotheses. The statistical significance testing results and the correction for multiple comparisons can be found in Appendix A.

## 5.1 Hypothesis **H1**

This evaluation is based on the outcome of the Sub-hypotheses **H1.1** and **H1.2**. Thus, we assess these two before giving the verdict on **H1**.

**H1.1** Before collating previous and new results, we look at the isolated findings from applying the ensemble methods. We hypothesized that there would be a set of systems that is the most accurate compared to any other combination. We can confirm this hypothesis by looking at the results produced in the experiments by using the ensemble methods. Due to the lack of a second system to combine TARGER with, no ensembles can be built to improve the relation prediction score $R$; thus, it is omitted.

Table 2 shows the results for each task. From it, we can see that the highest scores are unique numbers. This leads us to accept Sub-hypothesis **H1.1**.

**H1.2** This hypothesis compares the results from the benchmark and the *Vertical Integration* by opposing the best results from Table 1 and Table 2. For the sentence classification, the ensemble of TARGER, AURC, MARGOT, and TRABAM combined by stacking them (with logistic regression) slightly outperforms the previously most accurate single system TRABAM: $S = 0.842$ and $S = 0.832$, respectively. Statistical testing, however, reveals that the difference is not significant (cf. Appendix A). For the component identification where the two systems TRABAM and

TARGER were combined using the voting method ($C = 0.673$), they beat the previous best achieved by TRABAM ($C = 0.662$), with the difference being statistically significant. This is in contrast to the boundary detection, where the best ensemble result does not reach the most accurate single system: bagging TARGER, ArguminSci, MARGOT, and TRABAM scored $B = 0.497$, while TRABAM held the most accurate result in $B = 0.506$. As explained in the previous hypothesis, the relation prediction is omitted.

Since we found one of three ensembles to outperform single systems with a statistical significance, this leads us to accept Sub-hypothesis **H1.2**. Moreover, this indicates a correlation between the systems' errors since they do not seem to balance out their weaknesses in all cases. An exhaustive error analysis would be necessary to reveal more detailed insights.

**H1** We based the acceptance of Hypothesis **H1** on accepting both its corresponding sub-hypotheses, which we did as explained above. This means that we also accept Hypothesis **H1**.

## 5.2 Hypothesis **H2**

Table 3 shows the results of using the annotations produced by TRABAM (i.e., the most accurate system in the benchmark) as the input to subsequent steps for the other systems. The boldfaced numbers indicate improvements over the initial results with the system's own data. We can see that, except when combining TRABAM with TARGER for the component classification, we consistently outperform the benchmark results, and the differences are all statistically significant. Akin to the previous hypotheses, $R$ cannot be improved as we do not have another system to feed TRABAM's intermediate results into, or vice versa. Therefore, we also omit the relation prediction from evaluating this hypothesis. Since we could show that using more accurate intermediate results can improve the subsequent step of the AM pipeline, we accept Hypothesis **H2**.

### 5.3 Hypothesis H3

This hypothesis merges the *Vertical* and the *Horizontal Integration* into the *Combined Integration* to improve the accuracy for all tasks in the pipeline by also allowing intermediate data produced by ensemble methods. The results are shown in Table 4 with the boldfaced numbers indicating the tasks for which a new highest accuracy was achieved: $S = 0.842$ and $C = 0.673$ outperform the previous best single systems from BAM, but only the latter being statistically significantly different. This is in contrast to $B = 0.483$ and $R = 0.019$, where the former did not perform better, and the latter even lowered the score. Still, we have evidence that *Combined Integration* can be used to improve at least some tasks in the AM pipeline. Thus, we accept Hypothesis H3.

## 6 Limitations and Future Work

The major limitation of this work is that we implement post-hoc combinations. The reasons for not re-training the systems are two-fold. First, out of practicality to facilitate the addition of new AM systems and existing ensembles. Second, to set the scope of this research as opposed to works that look to explicitly fuse models such as neural networks by entangling the final classification layer such as described in Ribeiro et al. (2020). The latter opens up the future work of applying these techniques to the current five AM systems and mixing their latent representations, as opposed to only their outputs.

Another limitation is that all the included systems take a holistic view of the AM pipeline, and none is explicitly split into the four modules we infer for the ensemble methods. Given the success of (re-)combinations of components in other domains, this paper can, hence, be seen as *a call to action to systematically explore the effectiveness of functional components of the AM pipeline and share these for re-use by others*. Indeed, more broadly, the limited availability of AM systems and benchmark datasets hampered our ability to systematically compare a larger design space of system (component) combinations and limits the generalizability of our findings to other domains/datasets.

The plans for future efforts in this direction include two main points. As the next step, we aim to conduct an error analysis and explore the influences of the systems on the results. This will help identify the strengths and weaknesses of the individual systems and may provide insights about current AM systems' common weaknesses. Also, the new analysis can incorporate the spatial and temporal costs of the recombinations, which was omitted in this paper. In the future, should the number of argument mining systems considerably increase, the framework could be extended to include a predictor to choose the sets and sequences of argument miners for a given document that lead to an optimal accuracy improvement. This would involve developing a cost function.

## 7 Conclusions

This paper presented **DREAM**, a framework for the **D**eployment of **R**ecombination and **E**nsemble methods for **A**rgument **M**iners. Our work focuses on improving accuracy in Argument Mining (AM) and addresses the need for incremental improvements as opposed to current approaches, which tend to provide all-new solutions (Lawrence and Reed, 2019). With the **DREAM** framework, we implemented a flexible and automated approach for (re)combining AM systems. It offers the *Vertical Integration* (using ensemble methods for a single task), the *Horizontal Integration* (using different systems throughout the pipeline), and, finally, the *Combined Integration* (allowing sets of systems for the intermediate data).

Our findings confirmed the hypotheses formulated in this work. We showed that ensemble methods (Opitz and Maclin, 1999) could be used to improve accuracy for specific tasks in the AM pipeline. Furthermore, we demonstrated that recombination by using intermediate data from the most accurate system could lead to higher accuracy in the subsequent task. Finally, we highlighted the potential of deploying ensemble methods and recombination for AM. We hope this work will contribute to the further improvement of state-of-the-art and better generalizing AM systems across domains, a prevalent and well-acknowledged problem (Lippi and Torroni, 2016a).

## Acknowledgments

This research was partially funded by the Swiss National Science Foundation (SNSF) under Project "CrowdAlytics" (Grant Number 184994). The authors would also like to thank Luca Rossetto for his inputs and the anonymous reviewers for their constructive feedback.

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

# A  Appendix

## A.1  Statistical Significance Testing

We report the p-values where we claim to outperform a previously best result (indicated in bold in the original tables) in Table 5, Table 6, and Table 7. Bold numbers indicate statistically significant differences ($p < 0.05$), while * denotes small, ** medium, and *** large effect size. The testing for the statistical significance of the results was conducted with Autorank (Herbold, 2020). The detailed reports on the conducted tests for statistical significance, including all procedures and assumptions testing, are shown below.

**S for TARGER, AURC, MARGOT, and TRABAM (Stacking) vs. TRABAM**  The statistical analysis was conducted for 2 populations with 12 paired samples. The family-wise significance level of the tests is alpha=0.050. We failed to reject the null hypothesis that the population is normal for all populations (minimal observed p-value=0.328). Therefore, we assume that all populations are normal. No check for homogeneity was required because we only have two populations. Because we have only two populations and both populations are normal, we use the t-test to determine differences between the mean values of the populations and report the mean value (M) and the standard deviation (SD) for each population. We failed to reject the null hypothesis (p=0.180) of the paired t-test that the mean values of the populations C-trabam-test (M=0.834+-0.041, SD=0.054) and SB-S-targer+aurc+margot+trabam (M=0.838+-0.039, SD=0.052) are are equal. Therefore, *we assume that there is no statistically significant difference between the mean values of the populations.*

**C for TRABAM and TARGER (Voting) vs. TRABAM**  The statistical analysis was conducted for 2 populations with 12 paired samples. The family-wise significance level of the tests is alpha=0.050. We rejected the null hypothesis that the population is normal for the population C-V-trabam+targer (p=0.024). Therefore, we assume that not all populations are normal. No check for homogeneity was required because we only have two populations. Because we have only two populations and one of them is not normal, we use Wilcoxon's signed rank test to determine the differences in the central tendency and report the median (MD) and the median absolute deviation (MAD) for each population. We reject the null hypothesis (p=0.021) of Wilcoxon's signed rank test that population C-trabam-test (MD=0.671+-0.062, MAD=0.038) is not greater than population C-V-trabam+targer (MD=0.691+-0.048, MAD=0.017). Therefore, *we assume that the median of C-V-trabam+targer is significantly larger than the median value of C-trabam-test with a small effect size (gamma=-0.461).*

**B for TRABAM and AURC (Recombination) vs. AURC**  The statistical analysis was conducted for 2 populations with 12 paired samples. The family-wise significance level of the tests is alpha=0.050. We failed to reject the null hypothesis that the population is normal for all populations (minimal observed p-value=0.052). Therefore, we assume that all populations are normal. No check for homogeneity was required because we only have two populations. Because we have only two populations and both populations are normal, we use the t-test to determine differences between the mean values of the populations and report the mean value (M) and the standard deviation (SD) for each population. We reject the null hypothesis (p=0.000) of the paired t-test that the mean values of the populations B-aurc-test (M=0.028+-0.009, SD=0.012) and SB-R-(trabam)+(aurc) (M=0.488+-0.036, SD=0.047) are equal. Therefore, *we assume that the mean value of SB-R-(trabam)+(aurc) is significantly larger than the mean value of B-aurc-test with a large effect size (d=-13.252).*

**B for TRABAM and TARGER (Recombination) vs. TARGER**  The statistical analysis was conducted for 2 populations with 12 paired samples. The family-wise significance level of the tests is alpha=0.050. We failed to reject the null hypoth-

| Task | Method | Systems | | Previous best | p-value |
|:---:|---|---|---|---|---|
| S | Stacking | TARGER, AURC, MARGOT, TRABAM | | TRABAM | 0.180 |
| C | Voting | TRABAM, TARGER | | TRABAM | **0.021 *** |

Table 5: The p-values corresponding to the results reported in Table 2.

| From | To | B | C | R |
|---|---|---|---|---|
| TRABAM | AURC | **0.000 *** | - | - |
| TRABAM | TARGER | **0.018 *** | - | - |
| TRABAM | ArguminSci | **0.000 *** | **0.000 *** | - |
| TRABAM | MARGOT | **0.000 *** | **0.000 *** | - |

Table 6: The p-values corresponding to the results reported in Table 3.

esis that the population is normal for all populations (minimal observed p-value=0.189). Therefore, we assume that all populations are normal. No check for homogeneity was required because we only have two populations. Because we have only two populations and both populations are normal, we use the t-test to determine differences between the mean values of the populations and report the mean value (M) and the standard deviation (SD) for each population. We reject the null hypothesis (p=0.018) of the paired t-test that the mean values of the populations C-targer-test (M=0.485+-0.047, SD=0.063) and C-R-(trabam)+(targer) (M=0.504+-0.042, SD=0.056) are equal. Therefore, *we assume that the mean value of C-R-(trabam)+(targer) is significantly larger than the mean value of C-targer-test with a small effect size (d=-0.313).*

**B for TRABAM and ArguminSci (Recombination) vs. ArguminSci** The statistical analysis was conducted for 2 populations with 12 paired samples. The family-wise significance level of the tests is alpha=0.050. We failed to reject the null hypothesis that the population is normal for all populations (minimal observed p-value=0.120). Therefore, we assume that all populations are normal. No check for homogeneity was required because we only have two populations. Because we have only two populations and both populations are normal, we use the t-test to determine differences between the mean values of the populations and report the mean value (M) and the standard deviation (SD) for each population. We reject the null hypothesis (p=0.000) of the paired t-test that the mean values of the populations C-arguminsci-test (M=0.102+-0.013, SD=0.018) and C-R-(trabam)+(arguminsci) (M=0.287+-0.035, SD=0.047) are equal. There-

fore, *we assume that the mean value of C-R-(trabam)+(arguminsci) is significantly larger than the mean value of C-arguminsci-test with a large effect size (d=-5.227).*

**C for TRABAM and ArguminSci (Recombination) vs. ArguminSci** The statistical analysis was conducted for 2 populations with 12 paired samples. The family-wise significance level of the tests is alpha=0.050. We failed to reject the null hypothesis that the population is normal for all populations (minimal observed p-value=0.681). Therefore, we assume that all populations are normal. No check for homogeneity was required because we only have two populations. Because we have only two populations and both populations are normal, we use the t-test to determine differences between the mean values of the populations and report the mean value (M) and the standard deviation (SD) for each population. We reject the null hypothesis (p=0.000) of the paired t-test that the mean values of the populations C-arguminsci-test (M=0.093+-0.016, SD=0.021) and C-R-(trabam)+(arguminsci) (M=0.344+-0.040, SD=0.054) are equal. Therefore, *we assume that the mean value of C-R-(trabam)+(arguminsci) is significantly larger than the mean value of C-arguminsci-test with a large effect size (d=-6.140).*

**B for MARGOT and TRABAM (Recombination) vs. MARGOT** The statistical analysis was conducted for 2 populations with 12 paired samples. The family-wise significance level of the tests is alpha=0.050. We failed to reject the null hypothesis that the population is normal for all populations (minimal observed p-value=0.133). Therefore, we assume that all populations are normal. No check for homogeneity was required because we

| Task | Method | Systems | Previous best | p-value |
|------|--------|---------|---------------|---------|
| **S** | Stacking | TARGER, AURC, MARGOT, TRABAM | TRABAM | 0.180 |
| **C** | Voting | TRABAM, TARGER | TRABAM | **0.021 \*** |

Table 7: The p-values corresponding to the results reported in Table 4.

| Hypothesis | | | p-value | Rank | $(i/m)\alpha$ |
|------------|---|---|---------|------|---------------|
| B | B-aurc-test | SB-R-(trabam)+(aurc) | **1.933E-12** | 1 | 0.006 |
| C | C-arguminsci-test | C-R-(trabam)+(arguminsci) | **2.133E-09** | 2 | 0.013 |
| B | C-arguminsci-test | C-R-(trabam)+(arguminsci) | **1.866E-08** | 3 | 0.019 |
| B | C-margot-test | C-R-(trabam)+(margot) | **2.032E-07** | 4 | 0.025 |
| C | C-margot-test | C-R-(trabam)+(margot) | **4.266E-04** | 5 | 0.031 |
| B | C-targer-test | C-R-(trabam)+(targer) | **1.822E-02** | 6 | 0.038 |
| C | C-trabam-test | C-V-trabam+targer | **2.124E-02** | 7 | 0.044 |
| S | C-trabam-test | SB-S-targer+aurc+margot+trabam | 1.795E-01 | 8 | 0.050 |

Table 8: Calculations of the Benjamini-Hochberg procedure.

only have two populations. Because we have only two populations and both populations are normal, we use the t-test to determine differences between the mean values of the populations and report the mean value (M) and the standard deviation (SD) for each population. We reject the null hypothesis (p=0.000) of the paired t-test that the mean values of the populations C-margot-test (M=0.098+-0.014, SD=0.019) and C-R-(trabam)+(margot) (M=0.171+-0.020, SD=0.026) are equal. Therefore, we assume that the mean value of C-R-(trabam)+(margot) is significantly larger than the mean value of C-margot-test with a large effect size (d=-3.210).

**C for MARGOT and TRABAM (Recombination) vs. MARGOT** The statistical analysis was conducted for 2 populations with 12 paired samples. The family-wise significance level of the tests is alpha=0.050. We failed to reject the null hypothesis that the population is normal for all populations (minimal observed p-value=0.347). Therefore, we assume that all populations are normal. No check for homogeneity was required because we only have two populations. Because we have only two populations and both populations are normal, we use the t-test to determine differences between the mean values of the populations and report the mean value (M) and the standard deviation (SD) for each population. We reject the null hypothesis (p=0.000) of the paired t-test that the mean values of the populations C-margot-test (M=0.135+-0.031, SD=0.042) and C-R-(trabam)+(margot)

(M=0.164+-0.029, SD=0.039) are equal. Therefore, *we assume that the mean value of C-R-(trabam)+(margot) is significantly larger than the mean value of C-margot-test with a medium effect size (d=-0.743).*

**S for TARGER, AURC, MARGOT, and TRABAM (Stacking) vs. TRABAM** The statistical analysis was conducted for 2 populations with 12 paired samples. The family-wise significance level of the tests is alpha=0.050. We failed to reject the null hypothesis that the population is normal for all populations (minimal observed p-value=0.328). Therefore, we assume that all populations are normal. No check for homogeneity was required because we only have two populations. Because we have only two populations and both populations are normal, we use the t-test to determine differences between the mean values of the populations and report the mean value (M) and the standard deviation (SD) for each population. We failed to reject the null hypothesis (p=0.180) of the paired t-test that the mean values of the populations C-trabam-test (M=0.834+-0.041, SD=0.054) and SB-S-targer+aurc+margot+trabam (M=0.838+-0.039, SD=0.052) are are equal. Therefore, *we assume that there is no statistically significant difference between the mean values of the populations.*

**C for (TRABAM and TARGER (Voting)) and TRABAM (Recombination) vs. TRABAM** The statistical analysis was conducted for 2 populations with 12 paired samples. The family-wise significance level of the tests is alpha=0.050. We

rejected the null hypothesis that the population is normal for the population C-R-(trabam)+(C-V-trabam+targer) (p=0.024). Therefore, we assume that not all populations are normal. No check for homogeneity was required because we only have two populations. Because we have only two populations and one of them is not normal, we use Wilcoxon's signed rank test to determine the differences in the central tendency and report the median (MD) and the median absolute deviation (MAD) for each population. We reject the null hypothesis (p=0.021) of Wilcoxon's signed rank test that population C-trabam-test (MD=0.671+-0.062, MAD=0.038) is not greater than population C-R-(trabam)+(C-V-trabam+targer) (MD=0.691+-0.048, MAD=0.017). Therefore, *we assume that the median of C-R-(trabam)+(C-V-trabam+targer) is significantly larger than the median value of C-trabam-test with a small effect size (gamma=-0.461).*

## A.2 Multiple Comparisons Problem

With the results from the tests above, we correct for the multiple comparisons problem by using the Benjamini-Hochberg procedure (Benjamini and Hochberg, 1995) with a critical value of $\alpha = 0.05$. Table 8 shows the details of the calculations. From it, we see that all the differences are still statistically significant, even after correcting for the multiple comparisons problem.