# OpenReview forum: "DREAM: Deployment of Recombination and Ensembles in Argument Mining"
_EMNLP/2023/Conference — EMNLP 2023 Main_

### Official Review · Reviewer_9Pww · 2023-07-27

**Soundness:** 4

**Excitement:**

4: Strong: This paper deepens the understanding of some phenomenon or lowers the barriers to an existing research direction.

**Paper Topic And Main Contributions:**

The goal of this research is to provide a system for argument mining (AM) with ensemble methods, to test argument mining modules and allow for new combinations. The authors want to find the optimal combination of modules for the AM tasks and then test, whether this also produces the best result overall.

The methodology:
Use of 5 systems for AM
- Division of the AM pipeline into 4 tasks:
	- sentence classification: sentence tokenization and classification to argumentative/non-argumentative.
	- boundary detection: find the delineations of components in only the argumentative sentences
	- component identification: identify components whose boundaries have been detected
	- relation prediction: relies on the previously identified components to find the triples (subject and object are from the set of argumentative components) that constitute its output
- combine whole systems by giving a weight
- combine single modules by passing intermediate results to other systems



**Reasons To Accept:**

This is an interesting approach that allows modules to be recombined from components. This method is certainly transferable for other tasks.

Data and code are available on an anonymous GitHub account

**Reasons To Reject:**

The Related Work chapter refers to only one work. Perhaps one could introduce the components that are included in the evaluation here?
(This is currently in "Setup").

**Reproducibility:**

5: Could easily reproduce the results.

**Reviewer Confidence:**

4: Quite sure. I tried to check the important points carefully. It's unlikely, though conceivable, that I missed something that should affect my ratings.

---

> ### Author Rebuttal · Authors · 2023-08-28
>
> Dear Reviewer,
>
> Thank you very much for your valuable comments and very positive feedback. We would like to address your single concern, which poses a reason to reject. We had to keep the content of the paper to a minimum because of the page limit restriction. We can gladly extend the Related Work section with more elaborate explanations of the stages of the Argument Mining pipeline and the related components using the additional page for the camera-ready version.
> We expect to discuss a good dozen references in such a section. Note, however, that we are unaware of any other approach exploring the combination of existing approaches. So the related work would focus on different views of Argument Mining as a task, the existing general approaches to solve it, and an overview of “what works and what does not” regarding classifiers as well as features.
>
> Again, thank you very much for taking the time to review our paper. We hope that we addressed all your concerns that stood in the way of giving the paper a more positive rating and would greatly appreciate it if the updated score reflected this.

---

### Official Review · Reviewer_tpwx · 2023-07-30

**Soundness:** 4

**Excitement:**

3: Ambivalent: It has merits (e.g., it reports state-of-the-art results, the idea is nice), but there are key weaknesses (e.g., it describes incremental work), and it can significantly benefit from another round of revision. However, I won't object to accepting it if my co-reviewers champion it.

**Missing References:**

None (but this is not my area of research).

**Paper Topic And Main Contributions:**

This paper presents a framework for the use of ensemble approaches for various phases of he task of argument mining. The authors experiment with various ensemble setups and evaluate the results with existing data (BAM).

Code is available in a code repository.

**Questions For The Authors:**

A. What are the results of tests for statistical significance? - answered in the rebuttal, thanks

B. There is a possibility that the best results are just due to randomness (the Multiple Comparisons problem). What are the results after applying corrections such as Bonferroni correction? What are the results on a different test set? - answered in the rebuttal, thanks

**Reasons To Accept:**

A. The code is available for reproducibility.

B. Very good and detailed explanation of the approach.

C. A methodical approach to test the various ensemble configurations.

**Reasons To Reject:**

A. Ensemble approaches have been proven to improve over individual approaches in a wide range of tasks. The fact that it appears to be the case in this task as well is not surprising.

After the authors rebuttal, the following is no longer applicable:
B. It is unclear whether the conclusions are valid. It appears that the experiments are based on a single test set and therefore the results might suffer of the Multiple Comparisons problem: Given that multiple experiments have been made by exhausting all possible ensemble configurations, it is possible that the best results are due to a coincidence. The experiments did not include tests of statistical significance or statistical corrections to compensate for this.

**Reproducibility:**

5: Could easily reproduce the results.

**Reviewer Confidence:**

4: Quite sure. I tried to check the important points carefully. It's unlikely, though conceivable, that I missed something that should affect my ratings.

**Typos Grammar Style And Presentation Improvements:**

The quality of writing is very good and I did not find any typos or English errors.

---

> ### Author Rebuttal · Authors · 2023-08-28
>
> Dear Reviewer,
>
> Thank you very much for your valuable comments.
>
> We would like to rebut and answer the following concerns that you raised in your review in this particular order: (I) the status of ensemble methods in Argument Mining, (II a) the statistical significance testing and (II b) the multiple comparisons problem, (III) the lack of additional test sets, and (IV) the poor scientific soundness.
>
> (I) Ensemble Methods in Argument Mining
>
> We never claim that the results we obtained are surprising, but rather our goal is to improve the state-of-the-art and provide a framework to do so even further. To the best of our knowledge, we are the first to employ ensemble methods for the task of Argument Mining, also evidenced by the relatively short list of related works.
> Moreover, we see this paper as “a call to action to systematically explore the effectiveness of functional components of the AM pipeline and share these for re-use by others”, as emphasized in the Limitations and Future Work section.
>
> (II) Statistical Significance
>
> (a) Testing
>
> According to your comment, we performed the tests for statistical significance of the newly obtained results compared to the previous best runs from the BAM paper (Ruosch et al. 2022) using this benchmark. We report the p-values in tables corresponding to the ones in the DREAM paper. We find that, apart from one instance, all newly found best results are indeed statistically significantly different and, therefore, indeed valid. These calculations are to be added to the appendix of the camera-ready paper.
>
> \* denotes small effect size, ** medium, and *** large effect size; bold indicates statistically significant difference (p < 0.05)
> - Corresponding to Table 2
> | Task | Ensemble                    | Previous best | p-value |
> |------|-----------------------------|---------------|---------|
> | S    | S-targer+aurc+margot+trabam | TRABAM        | 0.180   |
> | C    | V-trabam+targer             | TRABAM        | **0.021** * |
> - Corresponding to Table 3
> | From   | To         | B         | C         | R |
> |--------|------------|-----------|-----------|---|
> | TRABAM | AURC       | **0.000** *** | -         | - |
> | TRABAM | TARGER     | **0.018** *   | -         | - |
> | TRABAM | ArguminSci | **0.000** *** | **0.000** *** | - |
> | TRABAM | MARGOT     | **0.000** *** | **0.000** **  | - |
> - Corresponding to Table 4
> | Task | Ensemble                    | Previous best | p-value |
> |------|-----------------------------|---------------|---------|
> | S    | S-targer+aurc+margot+trabam | TRABAM        | 0.180   |
> | C    | V-trabam+targer             | TRABAM        | **0.021** * |
>
> A detailed report on the conducted tests for statistical significance, including all procedures and assumptions testing, is included at the end of the rebuttal.
>
> (b) Multiple Comparisons Problem
>
> With the results from the tests above, we correct for the multiple comparisons problem by using the Benjamini-Hochberg procedure with a critical value of Q = 0.05. The table with the details of the calculations can be found below. From it, we see that all the differences are still statistically significant, even after correcting for the multiple comparisons problem. These calculations are to be added to the appendix of the camera-ready paper.
>
> Bold p-values in the table indicate that they are statistically significant after the correction.
> | Hypothesis                                     | p-value       | Rank | (i/m)Q |
> |------------------------------------------------|---------------|------|--------|
> | B B-aurc-test SB-R-(trabam)+(aurc)             | **1.933E-12** | 1    | 0.006  |
> | C C-arguminsci-test C-R-(trabam)+(arguminsci)  | **2.133E-09** | 2    | 0.013  |
> | B C-arguminsci-test C-R-(trabam)+(arguminsci)  | **1.866E-08** | 3    | 0.019  |
> | B C-margot-test C-R-(trabam)+(margot)          | **2.032E-07** | 4    | 0.025  |
> | C C-margot-test C-R-(trabam)+(margot)          | **4.266E-04** | 5    | 0.031  |
> | B C-targer-test C-R-(trabam)+(targer)          | **1.822E-02** | 6    | 0.038  |
> | C C-trabam-test C-V-trabam+targer              | **2.124E-02** | 7    | 0.044  |
> | S C-trabam-test SB-S-targer+aurc+margot+trabam | 1.795E-01     | 8    | 0.050  |
>
> (III) Lack of Additional Test Sets
>
> Unfortunately, including an additional test set would take more time than we have in the rebuttal period. We would need to obtain and pre-process a new data set, then train, execute, and evaluate all the systems as well as their combinations. Even when we only consider training the systems on new data, we are looking at several days of computing (see the run times of the systems in Table 1 of the BAM paper (Ruosch et al. 2022)). This is simply not feasible at the moment.
> As to why we only had one test set in the data to begin with lies in the fact that BAM is currently the only unifying benchmark in Argument Mining that produces comparable results. Its test data set only consists of one annotated corpus of scientific papers originally created by Lauscher et al. (2018) (Sci-Arg).  Note that we are unaware of any other dataset of scholarly papers annotated for Argument Mining. We do agree that the community urgently needs to get together and collect a suite of datasets – it is a crucial step in improving the performance of Argument Mining approaches.
>
> (IV) Poor Soundness
>
> We thank the reviewer for the very positive verdict given under “Reasons To Accept”, including the mention of a “methodical approach”. We understand that the poor rating of soundness has to do with the lack of statistical tests for the significance of the obtained results. We hope that the reviewer is satisfied with the deliberations and p-values given above and, therefore, is inclined to increase the appropriate score.
>
> Again, thank you very much for taking the time to review our paper. We hope that we addressed all your concerns that stood in the way of giving the paper a more positive rating and would greatly appreciate it if the updated score would reflect this.
>
> References
> * Florian Ruosch, Cristina Sarasua, and Abraham Bernstein. 2022. BAM: Benchmarking Argument Mining on Scientific Documents. In Proceedings of the Workshop on Scientific Document Understanding co-located with 36th AAAI Conference on Artificial Intelligence (AAAI 2022).
>
>
> Here is the detailed report on the conducted tests for statistical significance, including all procedures and assumptions testing:
>
> ************ S C-trabam-test SB-S-targer+aurc+margot+trabam ************
> The statistical analysis was conducted for 2 populations with 12 paired samples.
> The family-wise significance level of the tests is alpha=0.050.
> We failed to reject the null hypothesis that the population is normal for all populations (minimal observed p-value=0.328). Therefore, we assume that all populations are normal.
> No check for homogeneity was required because we only have two populations.
> Because we have only two populations and both populations are normal, we use the t-test to determine differences between the mean values of the populations and report the mean value (M)and the standard deviation (SD) for each population.
> We failed to reject the null hypothesis (p=0.180) of the paired t-test that the mean values of the populations C-trabam-test (M=0.834+-0.041, SD=0.054) and SB-S-targer+aurc+margot+trabam (M=0.838+-0.039, SD=0.052) are are equal. Therefore, we assume that there is no statistically significant difference between the mean values of the populations.
>
> ************ C C-trabam-test C-V-trabam+targer ************
> The statistical analysis was conducted for 2 populations with 12 paired samples.
> The family-wise significance level of the tests is alpha=0.050.
> We rejected the null hypothesis that the population is normal for the population C-V-trabam+targer (p=0.024). Therefore, we assume that not all populations are normal.
> No check for homogeneity was required because we only have two populations.
> Because we have only two populations and one of them is not normal, we use Wilcoxon's signed rank test to determine the differences in the central tendency and report the median (MD) and the median absolute deviation (MAD) for each population.
> We reject the null hypothesis (p=0.021) of Wilcoxon's signed rank test that population C-trabam-test (MD=0.671+-0.062, MAD=0.038) is not greater than population C-V-trabam+targer (MD=0.691+-0.048, MAD=0.017). Therefore, we assume that the median of C-V-trabam+targer is significantly larger than the median value of C-trabam-test with a small effect size (gamma=-0.461).
>
> ************ B B-aurc-test SB-R-(trabam)+(aurc) ************
> The statistical analysis was conducted for 2 populations with 12 paired samples.
> The family-wise significance level of the tests is alpha=0.050.
> We failed to reject the null hypothesis that the population is normal for all populations (minimal observed p-value=0.052). Therefore, we assume that all populations are normal.
> No check for homogeneity was required because we only have two populations.
> Because we have only two populations and both populations are normal, we use the t-test to determine differences between the mean values of the populations and report the mean value (M)and the standard deviation (SD) for each population.
> We reject the null hypothesis (p=0.000) of the paired t-test that the mean values of the populations B-aurc-test (M=0.028+-0.009, SD=0.012) and SB-R-(trabam)+(aurc) (M=0.488+-0.036, SD=0.047) are equal. Therefore, we assume that the mean value of SB-R-(trabam)+(aurc) is significantly larger than the mean value of B-aurc-test with a large effect size (d=-13.252).
>
> ************ B C-targer-test C-R-(trabam)+(targer) ************
> The statistical analysis was conducted for 2 populations with 12 paired samples.
> The family-wise significance level of the tests is alpha=0.050.
> We failed to reject the null hypothesis that the population is normal for all populations (minimal observed p-value=0.189). Therefore, we assume that all populations are normal.
> No check for homogeneity was required because we only have two populations.
> Because we have only two populations and both populations are normal, we use the t-test to determine differences between the mean values of the populations and report the mean value (M)and the standard deviation (SD) for each population.
> We reject the null hypothesis (p=0.018) of the paired t-test that the mean values of the populations C-targer-test (M=0.485+-0.047, SD=0.063) and C-R-(trabam)+(targer) (M=0.504+-0.042, SD=0.056) are equal. Therefore, we assume that the mean value of C-R-(trabam)+(targer) is significantly larger than the mean value of C-targer-test with a small effect size (d=-0.313).
>
> ************ B C-arguminsci-test C-R-(trabam)+(arguminsci) ************
> The statistical analysis was conducted for 2 populations with 12 paired samples.
> The family-wise significance level of the tests is alpha=0.050.
> We failed to reject the null hypothesis that the population is normal for all populations (minimal observed p-value=0.120). Therefore, we assume that all populations are normal.
> No check for homogeneity was required because we only have two populations.
> Because we have only two populations and both populations are normal, we use the t-test to determine differences between the mean values of the populations and report the mean value (M)and the standard deviation (SD) for each population.
> We reject the null hypothesis (p=0.000) of the paired t-test that the mean values of the populations C-arguminsci-test (M=0.102+-0.013, SD=0.018) and C-R-(trabam)+(arguminsci) (M=0.287+-0.035, SD=0.047) are equal. Therefore, we assume that the mean value of C-R-(trabam)+(arguminsci) is significantly larger than the mean value of C-arguminsci-test with a large effect size (d=-5.227).
>
> ************ C C-arguminsci-test C-R-(trabam)+(arguminsci) ************
> The statistical analysis was conducted for 2 populations with 12 paired samples.
> The family-wise significance level of the tests is alpha=0.050.
> We failed to reject the null hypothesis that the population is normal for all populations (minimal observed p-value=0.681). Therefore, we assume that all populations are normal.
> No check for homogeneity was required because we only have two populations.
> Because we have only two populations and both populations are normal, we use the t-test to determine differences between the mean values of the populations and report the mean value (M)and the standard deviation (SD) for each population.
> We reject the null hypothesis (p=0.000) of the paired t-test that the mean values of the populations C-arguminsci-test (M=0.093+-0.016, SD=0.021) and C-R-(trabam)+(arguminsci) (M=0.344+-0.040, SD=0.054) are equal. Therefore, we assume that the mean value of C-R-(trabam)+(arguminsci) is significantly larger than the mean value of C-arguminsci-test with a large effect size (d=-6.140).
>
> ************ B C-margot-test C-R-(trabam)+(margot) ************
> The statistical analysis was conducted for 2 populations with 12 paired samples.
> The family-wise significance level of the tests is alpha=0.050.
> We failed to reject the null hypothesis that the population is normal for all populations (minimal observed p-value=0.133). Therefore, we assume that all populations are normal.
> No check for homogeneity was required because we only have two populations.
> Because we have only two populations and both populations are normal, we use the t-test to determine differences between the mean values of the populations and report the mean value (M)and the standard deviation (SD) for each population.
> We reject the null hypothesis (p=0.000) of the paired t-test that the mean values of the populations C-margot-test (M=0.098+-0.014, SD=0.019) and C-R-(trabam)+(margot) (M=0.171+-0.020, SD=0.026) are equal. Therefore, we assume that the mean value of C-R-(trabam)+(margot) is significantly larger than the mean value of C-margot-test with a large effect size (d=-3.210).
>
> ************ C C-margot-test C-R-(trabam)+(margot) ************
> The statistical analysis was conducted for 2 populations with 12 paired samples.
> The family-wise significance level of the tests is alpha=0.050.
> We failed to reject the null hypothesis that the population is normal for all populations (minimal observed p-value=0.347). Therefore, we assume that all populations are normal.
> No check for homogeneity was required because we only have two populations.
> Because we have only two populations and both populations are normal, we use the t-test to determine differences between the mean values of the populations and report the mean value (M)and the standard deviation (SD) for each population.
> We reject the null hypothesis (p=0.000) of the paired t-test that the mean values of the populations C-margot-test (M=0.135+-0.031, SD=0.042) and C-R-(trabam)+(margot) (M=0.164+-0.029, SD=0.039) are equal. Therefore, we assume that the mean value of C-R-(trabam)+(margot) is significantly larger than the mean value of C-margot-test with a medium effect size (d=-0.743).
>
> ************ S C-trabam-test SB-S-targer+aurc+margot+trabam ************
> The statistical analysis was conducted for 2 populations with 12 paired samples.
> The family-wise significance level of the tests is alpha=0.050.
> We failed to reject the null hypothesis that the population is normal for all populations (minimal observed p-value=0.328). Therefore, we assume that all populations are normal.
> No check for homogeneity was required because we only have two populations.
> Because we have only two populations and both populations are normal, we use the t-test to determine differences between the mean values of the populations and report the mean value (M)and the standard deviation (SD) for each population.
> We failed to reject the null hypothesis (p=0.180) of the paired t-test that the mean values of the populations C-trabam-test (M=0.834+-0.041, SD=0.054) and SB-S-targer+aurc+margot+trabam (M=0.838+-0.039, SD=0.052) are are equal. Therefore, we assume that there is no statistically significant difference between the mean values of the populations.
>
> ************ C C-trabam-test C-R-(trabam)+(C-V-trabam+targer) ************
> The statistical analysis was conducted for 2 populations with 12 paired samples.
> The family-wise significance level of the tests is alpha=0.050.
> We rejected the null hypothesis that the population is normal for the population C-R-(trabam)+(C-V-trabam+targer) (p=0.024). Therefore, we assume that not all populations are normal.
> No check for homogeneity was required because we only have two populations.
> Because we have only two populations and one of them is not normal, we use Wilcoxon's signed rank test to determine the differences in the central tendency and report the median (MD) and the median absolute deviation (MAD) for each population.
> We reject the null hypothesis (p=0.021) of Wilcoxon's signed rank test that population C-trabam-test (MD=0.671+-0.062, MAD=0.038) is not greater than population C-R-(trabam)+(C-V-trabam+targer) (MD=0.691+-0.048, MAD=0.017). Therefore, we assume that the median of C-R-(trabam)+(C-V-trabam+targer) is significantly larger than the median value of C-trabam-test with a small effect size (gamma=-0.461).

---

### Official Review · Reviewer_1CYs · 2023-08-03

**Typos Grammar Style And Presentation Improvements:** Line 489
**Soundness:** 3

**Excitement:**

3: Ambivalent: It has merits (e.g., it reports state-of-the-art results, the idea is nice), but there are key weaknesses (e.g., it describes incremental work), and it can significantly benefit from another round of revision. However, I won't object to accepting it if my co-reviewers champion it.

**Paper Topic And Main Contributions:**

The authors introduce an Argument Mining (AM) framework that allows
for the automated combination of AM components in order to improve
accuracy for the tasks in the overall pipeline, the goal being
generalize well across domains by combining the strengths and
balance out the weaknesses.

**Questions For The Authors:**

A. Even from my lack of expert knowledge on the subject, a question
intrigues me. The authors claim that combining approaches in AM has
barely received any attention in previous literature. Is there any
reason for it? Does it have something to do with the reviewer's item A
in the previous section "Reasons to reject" ?

In relation to the authors' responses, the fact that the only costs to
consider are the adjustment and prediction of the combined methods
does not in any way detract from their estimation, given that The
practical interest of any computational process is determined by its
costs, both temporal and spatial. On the other hand, time costs cannot
be evaluated based on approximate estimates resulting from physical
timing (i.e. seconds), dependent on a multitude of external factors,
but rather based on formally defined time units. Even assuming that
the cost of training and running sets is several magnitudes less than
the cost of individual Argument Miners, the authors should make this
point as best they can.

Regarding the question of what could be the reasons why combining
approaches in AM has barely received any attention in previous
literature, the authors should reflect their response in the text.

**Reasons To Accept:**

I am not a specialist in the field, but precisely for that reason I
particularly appreciated the clarity of the writing and the (apparent)
rigor of the testing protocol. In this sense, the results of the
experimental tests that support the proposal have been convincing to
me, which is why I understand that the paper could have sufficient
quality to be accepted.

**Reasons To Reject:**

At no time do the authors estimate the computational and temporal
costs of their proposal, nor can they therefore compare them with
those of their competitors in the experimental tests. We are not talking about
a less important issue, but a determining one that can limit the
interest of the proposal to the theoretical framework.

**Reproducibility:**

3: Could reproduce the results with some difficulty. The settings of parameters are underspecified or subjectively determined; the training/evaluation data are not widely available.

**Reviewer Confidence:**

2: Willing to defend my evaluation, but it is fairly likely that I missed some details, didn't understand some central points, or can't be sure about the novelty of the work.

---

> ### Author Rebuttal · Authors · 2023-08-28
>
> Dear Reviewer,
>
> Thank you very much for your valuable comments.
>
> First, we would like to address your concern regarding the lack of reporting computational and temporal costs. Since we explicitly factored out the training and execution costs of the involved systems (see Section 4.1 Setup), we only consider the costs of combining them. Thus, the only costs to be considered are the fitting and prediction of the combination methods, and, therefore, no comparison can and should be made to the original systems. Given the magnitude of the data (the train and test data set contain approximately 242,000 tokens combined), the temporal costs for the ensemble methods were in the range of minutes (or even seconds). The computing time for all possible combinations (n = 309) for the Vertical Integration was 32 minutes and 35.16 seconds (resulting in an average of 6.33 seconds per), and for all results (n = 7) of the Horizontal Integration, it took 16.82 seconds (or 2.40 seconds on average). For the Combined Integration, no additional computations were necessary since it was merely a matter of selecting the best-performing combinations or single systems from the given results.
> We did not mention this in the paper because the cost to train and execute ensembles is several magnitudes smaller than the cost of the individual Argument Miners (seconds versus days, see the BAM paper by Ruosch et al. (2022)). To fully address your concern, we will gladly make use of the additional page for the camera-ready version and add a paragraph on all the computational costs (in terms of execution times).
>
> Second, to answer your question about combining approaches in AM barely receiving any attention. We do not have any evidence (apart from the lack of results when searching for these terms in literature) and can only speculate. One hint has been given in a very popular survey for the field of Argument Mining by Lawrence and Reed (2020): new publications tend to take a holistic view of the pipeline and, therefore, always present new approaches from scratch. This might have to do with the repeated advent of new techniques in Deep Learning and Large Language Models and people trying out those instead of focusing on improving existing methods. Nonetheless, if we take Ferrari et al. (2019) seriously, who showed that some advances of deep learning pipelines in the recommender systems domain may not always generalize well to other settings, and assume that the finding also holds for other domains then pursuing a recombination approach may be fruitful supporting our call to arms.
>
> Again, thank you very much for taking the time to review our paper. We hope that we addressed all your concerns that stood in the way of giving the paper a more positive rating and would greatly appreciate it if the updated score reflected this.
>
> References
> * Ferrari Dacrema, M., Cremonesi, P., & Jannach, D. (2019). Are we really making much progress? A worrying analysis of recent neural recommendation approaches. Proceedings of the 13th ACM Conference on Recommender Systems, 101–109.
> * John Lawrence and Chris Reed. 2019. Argument Mining: A Survey. Computational Linguistics, (August):1–55.
> * Florian Ruosch, Cristina Sarasua, and Abraham Bernstein. 2022. BAM: Benchmarking Argument Mining on Scientific Documents. In Proceedings of the Workshop on Scientific Document Understanding co-located with 36th AAAI Conference on Artificial Intelligence (AAAI 2022).

---

### Meta-Review · Area_Chair_qVYV · 2023-09-18

**Recommendation:** 3

**Metareview:**

This paper advances the state of the art in argument mining by showing that a pipelined approach, with ensembling, can outperform current end-to-end systems.

Reviewers were mostly ambivalent about this paper in terms of excitement, largely because the main contribution seems to be experimental results showing the advantages of a less-commonly used approach to this task. Reviewers also noted that improvements from using ensembles was not surprising or innovative, and that were results were only provided for a single dataset. However, they reported that the code to reproduce these experiments would be useful, and that these results would inform future work.

In terms of soundness, some initial concerns were raised, but reviewers seem largely satisfied with the authors' response, including an update that would correct for multiple comparisons. The one exception to this is Reviewer 1CYs's concerns about computational costs, but this seems like a somewhat secondary concern.

---

### Decision · Program_Chairs · 2023-10-07

**Decision:**

Accept-Main

**Comment:**

This paper advances the state of the art in argument mining by showing that a pipelined approach, with ensembling, can outperform current end-to-end systems.

Reviewers were mostly ambivalent about this paper in terms of excitement, largely because the main contribution seems to be experimental results showing the advantages of a less-commonly used approach to this task. Reviewers also noted that improvements from using ensembles was not surprising or innovative, and that were results were only provided for a single dataset. However, they reported that the code to reproduce these experiments would be useful, and that these results would inform future work.

In terms of soundness, some initial concerns were raised, but reviewers seem largely satisfied with the authors' response, including an update that would correct for multiple comparisons. The one exception to this is Reviewer 1CYs's concerns about computational costs, but this seems like a somewhat secondary concern.